# A cross-sectional study of fear of surgery in female breast cancer patients: Prevalence, severity, and sources, as well as relevant differences among patients experiencing high, moderate, and low fear of surgery

**Sophia Engel**[1]*, **Henrik Børsting Jacobsen**[1,2], **Silje Endresen Reme**[1,2]

**1** The Mind Body Lab, Department of Psychology, University of Oslo, Oslo, Norway, **2** Department of Pain Management and Research, Oslo University Hospital, Oslo, Norway

* a.s.engel@psykologi.uio.no

## Abstract

### Background

Fear of surgery has been associated with more postoperative pain, disability, and a lower quality of life among patients undergoing various surgical procedures. While qualitative studies indicate breast cancer patients to be afraid of surgery, detailed quantitative analyses are lacking. The present research aimed at investigating the prevalence, severity, and sources of fear of surgery in this patient group and to compare patients reporting different degrees of such fear.

### Methods

This cross-sectional study included 204 breast cancer patients, 18–70 years old, and scheduled for surgery at Oslo University Hospital, Norway. Following their preoperative visit participants completed validated psychological questionnaires online. Among these, the primary outcome measure, the Surgical Fear Questionnaire (SFQ; scores: 0–10 per item, 0–80 overall). Patients were grouped based on SFQ-percentiles (<25th = little, 25th–75th = moderate and >75th percentile = high fear) and compared on psychological (anxiety, depression, experienced injustice, optimism and expected postsurgical pain), sociodemographic, and medical outcomes.

### Results

195 patients completed the SFQ. On average fear of surgery was low (M = 26.41, SD = 16.0, median = 26, min-max = 0–80), but omnipresent. Only 1.5% (n = 3) indicated no fear at all. Overall, patients feared surgery itself the most (M = 3.64, SD = 2.8). Groups differed significantly (p < .001) in their experience of anxiety, depression, and injustice, as well as their disposition to be optimistic, and expectance of postsurgical pain. Differences between groups concerning demographic and medical information were largely insignificant.

**Data Availability Statement:** As soon as the datafile is anonymized, it will be available through the Norwegian Centre for Research Data (NSD)

Archive where it will be stored long-term (contact: https://www.nsd.no/en/find-data). Through this portal, researchers can apply for access to the data. NSD will in collaboration with our data access committee be responsible for reviewing requests and providing access to the anonymous data. The Data Access Committee includes Silje Endresen Reme (siljerem@uio.no), Henrik Børsting Jacobsen (h.b.jacobsen@psykologi.uio.no), Lars-Petter Granan (largra@ous-hf.no) and Tone Marte Ljoså (Tone.Marte.Ljosaa@usn.no).

**Funding:** The PREVENT trial is funded by a grant received by SER from the Norwegian Cancer Society (grant number: 201906-2019; https://kreftforeningen.no/forskning/). The funders had and will not have a role in study design, data collection and analysis, decision to publish, or preparation of the manuscript.

**Competing interests:** The authors have declared that no competing interests exist.

## Discussion

This study was the first to demonstrate fear of surgery to be prevalent and relevant among female breast cancer patients. The higher a patients' fear group, the poorer their preoperative psychological constitution. This, largely irrespective of their current diagnoses or treatments, medical history, and demographics. Fear of surgery might thus cater as a prognostic marker and treatment target in this patient group. However, given the cross-sectional character of the present data, prognostic studies are needed to evaluate such claims.

## Introduction

Being diagnosed with breast cancer (BC) constitutes a major stressor, for some even traumatic experience, prompting emotions like fear, anxiety, and distress in affected individuals [1, 2]. Worldwide, more than two million people, mostly females (> 99%), are diagnosed with BC every year, making it the most prevalent cancer overall [3, 4]. Surgery constitutes a major step in treatment and almost all patients are subjected to it [5]. Prior evidence shows that levels of anxiety among this patient group are highest in the presurgical phase [6–9]. Importantly, preoperative anxiety has been associated with many adverse pre-, peri- and postsurgical outcomes in female BC patients [10] and other surgical populations [11–18]. One factor contributing to preoperative anxiety is fear of surgery [19, 20]. Fear of surgery comprises fears patients experience with respect to short- (e.g., surgery, anesthesia) and/or long-term (e.g., health deterioration, incomplete recovery) aspects of a surgical intervention they are about to undergo [20]. Although still limited, prior evidence indicates BC patients are afraid of their impending surgery [21–23]. In a qualitative interview study it was shown that patients feared various aspects of the operation (pain, postoperative nausea, being dependent on the help of others following surgery), as well as the anesthesia (losing control) [22]. Moreover, according to another study in the frame of which structural clinical interviews were conducted with BC patients who had recently undergone surgery, 11.8% would even describe their fear of mastectomy as traumatic [1]. Complementing these results are those of a prospective study which assessed fear of surgery quantitatively by means of a single 4-point Likert scale ('not at all' to 'very much') in a sample of 761 females with unilateral primary non-metastatic BC. In this study, patients were generally afraid of surgery and those with poor preoperative mental health were described as experiencing more intense fears [21]. As 'poor mental health', however, was not further specified in the article, it remains difficult to interpret this finding. Lastly, a case-control study found that for 27.3% of all treatment-refusing patients with suspected or diagnosed BC who stated a reason for refusal, that reason was fear of surgery [23]. Taken together, according to qualitative and limited quantitative evidence, BC patients appear to be afraid of their surgical treatment and such fear might even affect their prognosis.

Fear of surgery can be measured by the validated Surgical Fear Questionnaire (SFQ) [20]. Despite a widespread use in various elective surgery populations, so far only one study reported SFQ-scores of female BC patients. The SFQ, however, was not a primary outcome in that study, and so far, the results are only described in a doctoral dissertation, but not yet published in a peer-reviewed journal [24]. Finally, an interpretation of reported mean SFQ-values of this study remains difficult as these were also not discussed in the context of prior studies [24]. Among in- and outpatients undergoing various surgical procedures, fears about short-term aspects of the upcoming surgery were shown to predict acute pain up until four days postoperatively [25, 26]. Moreover, fears such patients had about long-term aspects of the

surgery predicted chronic postsurgical pain, disability and a less improved quality of life up until one year postoperatively [25, 27–30]. Lastly, results from a study of cataract surgery show that patients who would appreciate informational, psychological, relaxational and/or pharmacological presurgical treatment targeted at reducing fear of surgery exhibited significantly higher SFQ-scores than those who wouldn't [31]. In summary, fear of surgery affects certain short- and long-term postoperative outcomes and among patients scoring high on the SFQ, there appears to be a call for tailored presurgical treatment aimed at reducing such fears.

Until today, detailed quantitative analyses of the prevalence, severity, and sources of fear of surgery in a naturalistic sample of female BC patients are lacking. The objective of the present study was to address this paucity by implementing the SFQ to cross-sectionally measure fear of surgery in a representative sample of BC patients, as well as to compare patients experiencing varying levels of fear of surgery on a range of targeted sociodemographic and clinical outcomes [32]. Based on results from prior research on fear of surgery in BC patients, it was hypothesized that most patients would experience moderate fear of surgery (SFQ $\geq$ 35) and that the surgery and the anesthesia would be the most feared aspects [21, 22]. Further, it was hypothesized that fear related to short- instead of long-term aspects of the surgery would be of greater magnitude [25, 27]. Outcomes for the planned comparisons were chosen based on their theoretical or priorly established potential to affect or be related to fear of surgery. It was hypothesized, that patients exhibiting higher fear would be younger, not married, scheduled for mastectomy, instead of breast-conserving therapy, scoring poorer on pain measures and psychological outcomes, as well as more likely to consume antidepressant medication [1, 6, 21, 25–31].

## Methods

### Setting and participants

The current study is based on cross-sectional data obtained from the PREVENT trial, a randomized controlled trial conducted by the 'Mind Body Lab' of the Department of Psychology at the University of Oslo, Norway [32]. The overall aim of the trial was to investigate the effectiveness of combined preoperative hypnosis and postoperative interned-based Acceptance-Commitment-Therapy on the prevention of postsurgical pain and fatigue in BC patients. To this end, 205 women, 18–70 years old, diagnosed with BC (including non-/invasive, primary/recurrent, metastatic, uni-/bilateral, genetic) and awaiting surgical treatment at the Department for Breast and Endocrine Surgery at Oslo University Hospital were included in the period from October 2020 to March 2022. Exclusion criteria were insufficient spoken or written Norwegian, an inability to fill out questionnaires, cognitive or psychiatric impairment, and having other malignancies.

Data presented in the present manuscript stem from baseline measurements of the PREVENT trial. These measurements were conducted in the time between women's preoperative meeting and their surgery. After a BC diagnosis was confirmed, eligible patients received oral and written information about the study. Interested patients were scheduled for a meeting with the trials' study nurse for the day of the preoperative assessment. Preceding inclusion, patients declared their voluntary participation and gave written informed consent.

Data was collected electronically using Viedoc™, the approved data management software for clinical trials at Oslo University Hospital. The study nurse introduced participants to their personal 'VieDocMe-account' and asked them to fill in the baseline questionnaires before their surgery. The questionnaire selection of the PREVENT trial was reviewed by a patient representative before it was finalized. The representative estimated that it would take 15–20 minutes to

complete the baseline questionnaires and regarded the associated burden as within reasonable limits for the patient population at hand [32].

## Outcome measures

**Primary outcome measure.** The primary outcome, fear of surgery, was assessed by means of the SFQ. Reliability and validity of the instrument have been demonstrated across various adult surgical populations [20]. The questionnaire consists of 8 items written in the first person ('I am afraid . . .') and scored on an 11-point scale (0 'not at all afraid' to 10 'very afraid', range 0–80) [20]. Items can be further divided into 2 subscales (SFQ-S and SFQ-L) consisting of 4 items each (range 0–40). The SFQ-S assesses fear of short-term aspects of the surgery (surgical procedure, anesthesia, postoperative pain and/or side-effects) and the SFQ-L fear of long-term aspects (health deterioration, failing of operation, incomplete recovery, and/or long duration of rehabilitation). Higher total scores indicate more fear. For the PREVENT trial the English version of the SFQ was first translated to Norwegian using the forward-backward method and then validated (see S1 File).

**Secondary outcome measures.** *Sociodemographic variables*. Sociodemographic information involved age, education, marital status, parenthood, and occupational status.

*Medical history*. With respect to participants' medical history prior psychiatric diagnoses and current usage of antipsychotics and/or antidepressants and/or benzodiazepines were extracted from patient charts as a proxy of psychological vulnerabilities. Further, participants detailed previous (breast) cancer diagnoses and surgical experiences in their VieDocMe-account.

*Current diagnosis and treatment*. Concerning patients' current diagnosis and treatment the following information was retrieved from medical records: Exact BC diagnosis including metastatic status, scheduled surgical procedure for the breast (breast-conserving surgery or mastectomy with/without reconstruction) and the axilla (sentinel lymph node biopsy, axillary lymph node dissection), planned hospital stay postoperatively.

*Pain*. Preexisting pain was investigated using 10 questions of a modified version of the Fibromyalgia Survey Diagnostic-2016 criteria [33], developed to assess widespread chronic pain and translated and validated for Norwegian-speaking individuals [34]. Five items inquired whether participants experienced pain in any of the four body quadrants or their neck or back. The other five asked whether such pain had persisted for more than three months. An affirmative answer to any of the latter five was counted as chronic pain. An affirmation to one of the first, but not last five questions was counted as sporadic pain.

Expected postsurgical pain was assessed by a numeric rating scale ranging from 0 'no pain' to 10 'worst imaginable pain'. A comparable single-item approach has been shown to be predictive for acute postsurgical pain previously [35]. Based on validated cutoffs for the assessment of pain intensity scores $\geq 7$ were interpreted as expecting severe pain [36].

*Psychological outcome measures*. Anxiety and depression were assessed by means of the reliable and valid Norwegian version of the Hospital Anxiety and Depression Scale [37–39]. The scale consists of 14 items, 7 relating to anxiety and depression each. Items are rated on a 4-point scale ranging from 0–3 (range per subscale: 0–21). Clinically relevant anxiety and depression were shown to be predicted with an optimal trade-off between sensitivity and specificity by scores of 8 and 7 respectively [39, 40].

Dispositional optimism was investigated using the validated Norwegian version of the Life Orientation Test-Revised [41]. The questionnaire asks respondents to indicate their agreement to 10 statements on a 5-point scale (0 'strongly disagree' to 4 'strongly agree'). Of these 3 assess optimism, 3 pessimism and 4 are filler items. Upon scoring, pessimism-items were reverse

coded, and filler-items left out as advised by the scale developers [42]. Possible total scores range from 0–24 with higher scores indicating more optimism.

The degree to which patients experienced irreparability, unfairness and/or blame with respect to their current health condition was measured using the Injustice Experience Questionnaire [43]. To reduce the burden on participants, a shorter version containing only 5 of originally 12 items was used. Adequate scale validity of the abbreviated instrument was demonstrated previously [32]. Items are scored on 5-point scales (0 'never' to 4 'always', range 0–20) and higher scores indicate more experienced injustice. To adapt the scale to a BC population, the wording was slightly adjusted.

Perceived social support was assessed by 5 questions inquiring to which degree patients experienced support from their partner, family, friends, colleagues, and neighbors to be available to them. The original questions stem from Skovbjerg et al. [44]. For the PREVENT trial, one additional item asking about the support of 'others' was added. Items are rated on 6-point scales (1 'always' to 6 'never', range 0–36). Scores were reverse coded so that higher scores indicate more support.

Cronbach's alpha was found to be $\geq 0.8$ for all questionnaires employed in the present study indicating adequate internal consistency.

## Statistical analyses

Statistical analyses were performed using 'R' (R3.6.0; for the complete syntax see S2 File). Results were considered statistically significant at p < .05.

The overall percentage of missing values was 0.76%, which is within a range asserted to be inconsequential [45, 46]. Moreover, Little's Missing Completely At Random (MCAR) Test proved data to be MCAR ($\chi^2(1522) = 1547,34$, p = .320) [47]. Patients with 100% missing data on the primary outcome measure (n = 9) were excluded from further analyses (see Fig 1).

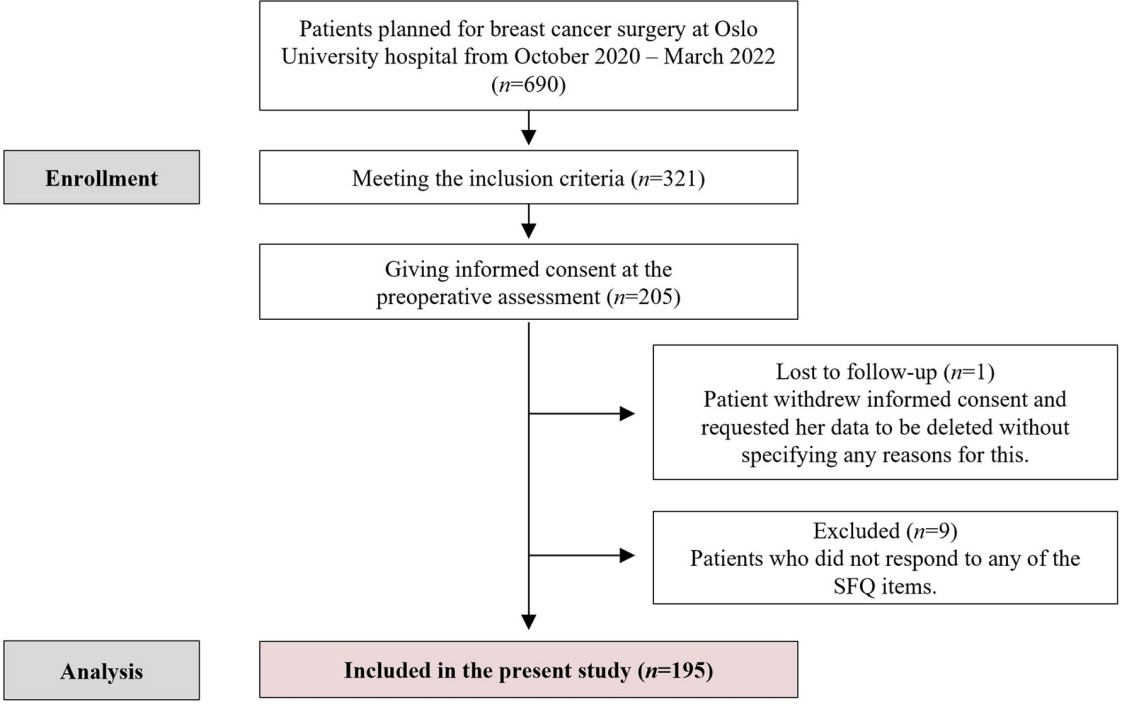

**Fig 1. Flowchart of the inclusion process.** Abbreviations: SFQ, Surgical Fear Questionnaire.

To optimize power of planned analyses, multiple imputation was employed (100 imputations with 20 iterations each [48]). Density and convergence plots revealed a good fit of the imputation model.

Descriptive statistics were analyzed as means (M) and standard deviations (SD) for continuous and frequencies (N) and percentages (%) for categorical variables.

The prevalence and severity of fear of surgery was assessed by means of descriptive statistics of the SFQ-total score. Sources of fear were investigated using descriptive statistics for the SFQ-subscales and individual items. A one-tailed paired samples t-test was employed to compare short- and long-term fears.

Validated cutoffs for the SFQ have not been described before. Consequently, to enable comparisons among participants experiencing differing levels of fear of surgery, meaningful SFQ-cutoffs needed to be chosen. Given the cross-sectional design, the usage of a receiver-operator-criterion-curve was discarded. Further, since the density plot of SFQ-Total scores clearly showed three groups, initial plans of performing a Median split were rejected as well. Since the borders of observed groups were close to the 25th and 75th percentiles (scores of 14 and 36 respectively) it was decided to use these values as grouping criteria (see Fig 2). Also, from a theoretical perspective it seemed plausible to claim that the 25% lowest and highest scores would indicate low and high fear respectively and that anything in between would represent moderate fear of surgery. Additionally, ANOVA-tests confirmed the difference in SFQ-total scores

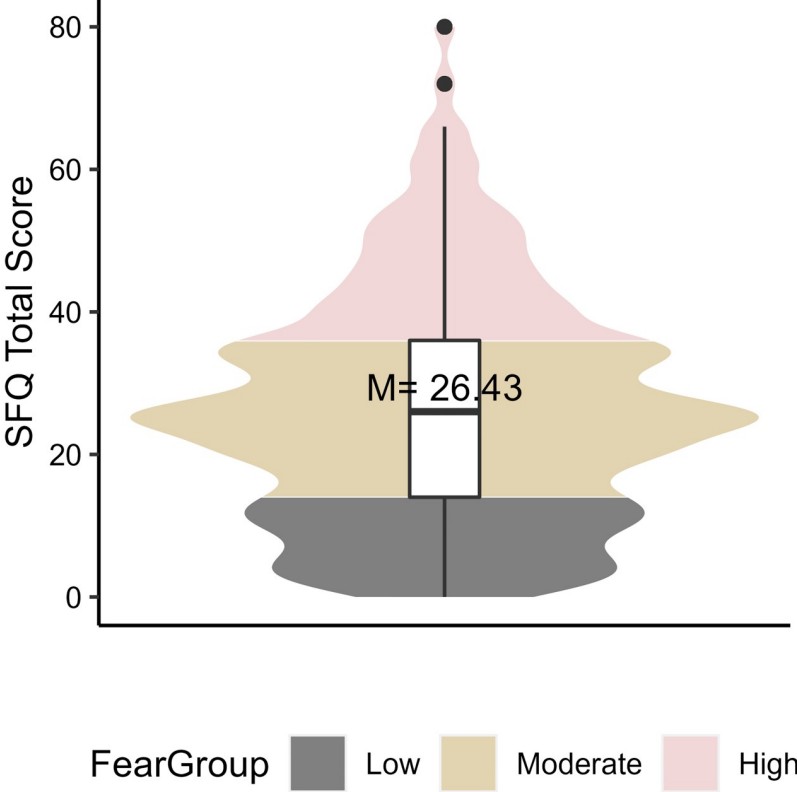

**Fig 2. Violin plot of total scores on the Surgical Fear Questionnaire (SFQ).** The shape of the figure represents a density plot. Wider regions indicate more frequent values. In the middle a boxplot is displayed. Whiskers illustrate the total spread of scores and the box the interquartile range (IQR). The median (= 26) is indicated by a black line. Statistical outliers (1.5xIQR) are indicated by black dots. The three fear groups are represented by different colors: < 25th percentile = low fear = grey, 25th to 75th percentile = moderate fear = beige, >75th percentile = high fear = pink.

between the three groups to be significant ($F(2,192) = 417,19$, p = < .001, see S1 Table), lending further justification for chosen cutoffs.

The three fear groups were compared on all socio-demographic, medical and psychological outcomes described above. ANOVA-tests were employed for continuous and chi-square-tests of independence for categorical outcomes. As groups were large ($\geq 25$), ANOVAs could be assumed to be robust against non-normality. Homoscedasticity was tested using Levene's test. If significant, the Welch-F, instead of ANOVA-F, ratio was interpreted. Post-hoc comparisons were completed using Tukey's HSD when homoscedasticity could be assumed and the Games-Howell procedure when this was not the case. For the chi-squares the Pearson-coefficient was interpreted when all expected counts were $\geq 5$ and Fisher's exact when this was not given. In both cases, post-hoc comparisons were completed using the Bonferroni-method. To reduce the risk of a Type I error due to multiple testing the Benjamini-Hochberg correction was applied [49].

As an explorative analysis, the overlap between high fear of surgery and psychological outcomes with defined clinical cutoffs (anxiety, depression, expected postsurgical pain) was investigated by means of Venn diagrams.

Finally, two sensitivity analyses were performed. Excluded patients, as well as those who declined participation or dropped out of the study, were compared to the final sample by means of independent samples t-tests and chi-square tests. Further, due to the large spread of scores in the high fear group and the fact that based on a purely visual evaluation of the density curve of SFQ total scores also a higher cutoff could have been chosen for this group, all comparisons among groups were repeated using a cutoff of 44 (instead of 36) to mark the transition from moderate to high fear.

### Ethical considerations

The current study, as well as the PREVENT trial at large, comply with good clinical practice and the latest version of the declaration of Helsinki. The PREVENT trial has been approved by the Regional Ethical Committee (reference number 67725), the Data Protection Officer at Oslo University Hospital (reference number 20/00831), the Norwegian Social Science Data Services (project number: 219790) and is registered at Clinicaltrials.gov (NTC04518085). Institutional guidelines concerning storing and publishing sensitive data were followed.

## Results

### Description of the sample

Of the 205 patients sampled for the PREVENT trial, 195 were included in the present study. The patient flow is detailed further in Fig 1. Women in the final sample were on average 54 years old (SD = 9.75, range 32–70). Most had a partner (67,7%, n = 132) and children (83.0%, n = 162) and were working (79.0%, n = 154). Few had been diagnosed with another cancer in the past (12.8%, n = 25). Many had undergone surgery priorly; 80% (n = 156) non-breast and 17.4% (n = 34) breast surgery. The large majority was diagnosed with primary invasive BC (85.07%, n = 166). This was followed by primary noninvasive (10.77% (n = 21)), recurrent (2.56% (n = 5)), metastasized primary (0.57% (n = 1)) and metastasized recurrent BC (1.03% (n = 2)). Most patients were scheduled for breast conserving therapy (68.07%, n = 133) and a sentinel node procedure (82.6%, n = 161).

### Prevalence and severity of fear of surgery

Almost all patients (98.5%, n = 192) exhibited fear of surgery to some degree (SFQ-score $\geq 0$) and the full range of possible total scores (0–80) was observed. The mean and median SFQ

total score were 26.43 (SD = 15.99) and 26, respectively. A statistical summary of SFQ total scores is displayed in Fig 2.

## Sources of fear of surgery

The dispersion of fear scores per SFQ subscale is displayed in Fig 3. On average, patients were more afraid of the short-, compared to long-term, aspects of the surgery (mean difference = 0.52, standard error = .56). This difference, however, was not statistically significant ($t$ ($193.98$) = 0.948, $p$ .172)). Of all eight potential fear sources investigated, the three most feared were the surgery (M(SD) = 3.64(2.76)), postoperative health deterioration (M(SD) = 3.61 (2.73)) and pain (M(SD) = 3.57(2.33)). This was followed by fear of side-effects of the surgery (M(SD) = 3.29(2.58)), a long rehabilitation (M(SD) = 3.25(2.44)), failing of the surgery (M (SD) = 3.20(2.87)), and the anaesthesia (M(SD) = 2.97(2.95)). An incomplete postoperative recovery was the item of the SFQ the participants feared the least (M(SD) = 2.90(2.49)).

## Description and comparison of fear groups

Table 1 details descriptive data and corresponding statistical analyses comparing the three fear groups (high, moderate, low) on all investigated outcomes. Most important results were as follows: Regarding sociodemographic factors, groups differed only in age. Patients with low, compared to moderate fear were significantly younger. Concerning prior medical history,

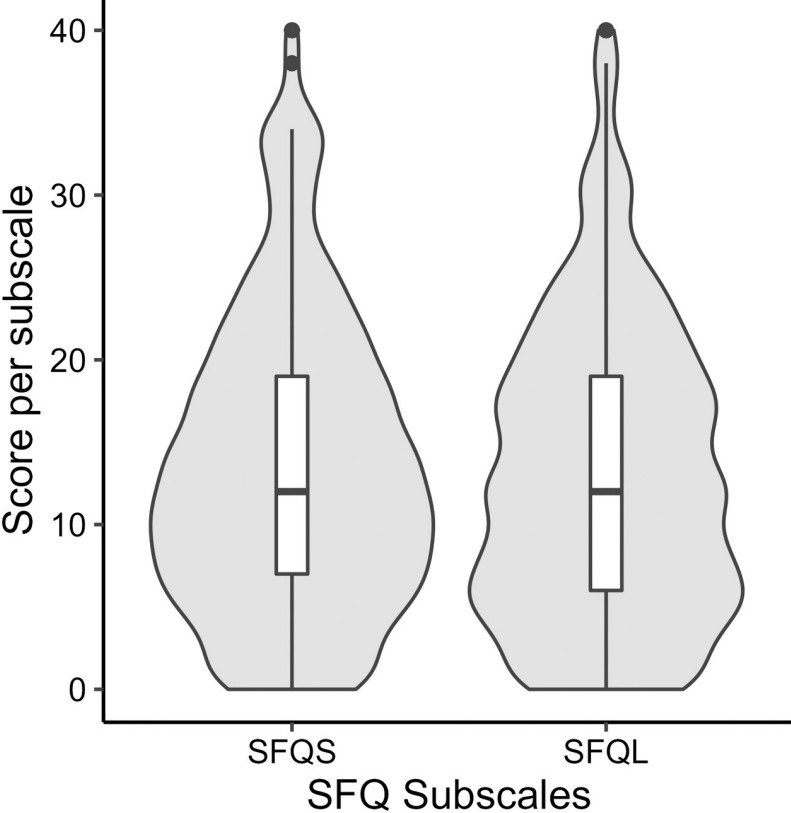

**Fig 3. Violin plots of total scores on the subscales, short- and long-term fears, of the Surgical Fear Questionnaire (SFQ-S & SFQ-L).** For both scales the full range of possible scores was achieved (0–40). Means were similar (SFQ-S, M (SD) = 13.47(8.86); SFQ-L, M(SD) = 12.95(8.90)).

**Table 1. Comparisons between the three fear groups (high, moderate, low) with respect to targeted outcomes.**

| | Fear | | | ANOVA/Chi-square | | Post-hoc Analyses | | |
|---|---|---|---|---|---|---|---|---|
| | High (≥36) N = 50 | Moderate (15–35) N = 98 | Low (≤14) N = 47 | | | | | |
| | M(SD)/N (%) | M(SD)/N(%) | M(SD)/N (%) | $F(df, df)/\chi^2(df, N)$ | p | High vs. Low | High vs. Moderate | Moderate vs. Low |
| **Sociodemographic variables** | | | | | | | | |
| Age | 52.63(9.61) | 52.45(10.10) | 56.96(8.46) | $F(2,192) = 3.87$ | 0.083 | .097 | .961 | .024 |
| Marital status (partner or married) | 31(62.00%) | 70(71.43%) | 31(65.96%) | $\chi^2(2,N = 195) = 1.43$ | 0.718 | | | |
| Children | 42(84.00%) | 82(83.49%) | 38(80.85%) | $\chi^2(2,N = 195) = 0.21$ | 0.937 | | | |
| Education (higher or university) | 37(74.00%) | 72(73.66%) | 38(80.85%) | $\chi^2(2,N = 195) = 0.98$ | 0.764 | | | |
| Working | 11(21.02%) | 21(21.43%) | 9(19.15%) | $\chi^2(2,N = 195) = 0.13$ | 0.937 | | | |
| **Medical history** | | | | | | | | |
| Cancer | 4(8.00%) | 16(16.33%) | 5(10.64%) | $\chi^2(2,N = 195) = 2.32$ | 0.595 | | | |
| Psychiatric diagnoses | 4(7.22%) | 5(5.10%) | 1(2.13%) | $\chi^2(2,N = 195) = 1.41$ | 0.718 | | | |
| Use of antidepressants, antipsychotics and/or benzodiazepines | 8(16.00%) | 15(15.31%) | 6(12.77%) | $\chi^2(2,N = 195) = 0.23$ | 0.937 | | | |
| Number of surgeries | 2.18(2.34) | 1.83(1.58) | 2.91(2.52) | $F_{Welch}(2,84.60) = 0.07$ | 0.083 | .301 | .602 | .023 |
| Breast surgery | 2(4.00%) | 24(24.49%) | 8(17.02%) | $\chi^2(2,N = 195) = 9.66$ | 0.033 | | ≤.05 | |
| **Current diagnosis and treatment** | | | | | | | | |
| Primary non-invasive BC | 7(14.00%) | 8(8.16%) | 6(12.77%) | $\chi^2(2,N = 195) = 1.21$ | 0.718 | | | |
| Primary invasive BC | 42(83.76%) | 84(85.71%) | 40(85.11%) | | | | | |
| Primary BC + Metastases | (0.00%) | 0(0.00%) | 1(2.13%) | | | | | |
| Recurrent BC | 1(2.00%) | 4(4.08%) | 0(0.00%) | | | | | |
| Recurrent BC + Metastases | 0 (0.00%) | 2(2.04%) | 0(0.00%) | | | | | |
| Neoadjuvant treatment | 17(33.22%) | 32(32.70%) | 17(36.17%) | $\chi^2(2,195) = 0.18$ | 0.937 | | | |
| Mastectomy with/without primary reconstruction | 18(36.0%) | 29(29.60%) | 11(23.40%) | $\chi^2(2,195) = 2.31$ | 00.595 | | | |
| Breast-conserving surgery with/without oncoplasty | 31(62.00%) | 65(66.30%) | 36(76.60%) | $\chi^2(2,195) = 2.20$ | 0.595 | | | |
| SN | 38(76.98%) | 83(84.69%) | 39(82.98%) | $\chi^2(2,195) = 1.26$ | 0.718 | | | |
| ALND | 11(21.12%) | 14(14.29%) | 5(10.64%) | $\chi^2(2,195) = 2.24$ | 0.595 | | | |
| Hospital | 37(73.26%) | 60(61.22%) | 25(53.19%) | $\chi^2(8,N = 195) = 4.38$ | 0.318 | | | |
| **Pain** | | | | | | | | |
| Sporadic | 5(10.00%) | 14(14.29%) | 4(8.51%) | $\chi^2(2,N = 195) = 1.49$ | 0.718 | | | |
| Chronic | 21(42.00%) | 35(35.71%) | 20(42.55%) | $\chi^2(2,N = 195) = 0.88$ | 0.765 | | | |
| Expected postsurgical pain | 6.11(1.85) | 4.73(1.81) | 3.87(1.9) | $F(2,192) = 18.608$ | < .001 | < .001 | < .001 | .026 |
| Expecting severe postsurgical pain* | 20(40.00%) | 15(15.31%) | 2(4.26%) | | | | | |
| **Psychological measures** | | | | | | | | |
| HADS-A—overall | 9.18(3.72) | 7.19(3.83) | 3.69(3.25) | $F(2,192) = 27.89$ | < .001 | < .001 | .006 | < .001 |
| Anxiety symptoms** | 35(70.00%) | 38(38.78%) | 6(12.77%) | | | | | |
| HADS-D | 4.65(2.94) | 3.61(2.76) | 1.75(1.96) | $F_{Welch}(2,107.01) = 19.63$ | < .001 | < .001 | .102 | < .001 |
| Depression symptoms*** | 13(26.00%) | 17(17.35%) | 3(6.38%) | | | | | |
| IEQ | 8.34(4.19) | 6.44(3.99) | 2.66(2.78) | $F_{Welch}(2,107.55) = 38.06$ | < .001 | < .001 | .025 | < .001 |
| LOT-R | 15.45(3.53) | 17.08(3.94) | 19.02(3.85) | $F(2,192) = 10.59$ | < .001 | < .001 | .030 | .016 |

*(Continued)*

**Table 1.** (Continued)

| | Fear | | | | | Post-hoc Analyses | | |
|---|---|---|---|---|---|---|---|---|
| | High (≥36) N = 50 | Moderate (15–35) N = 98 | Low (≤14) N = 47 | ANOVA/Chi-square | | High vs. Low | High vs. Moderate | Moderate vs. Low |
| | M(SD)/N (%) | M(SD)/N(%) | M(SD)/N (%) | $F(df, df)/\chi^2(df, N)$ | p | | | |
| Perceived Social Support | 25.72(5.77) | 25.68(5.86) | 27.44(5.68) | $F(2,192) = 1.77$ | .433 | .291 | .997 | .173 |

Abbreviations

vs, versus; HADS-A/D, Hospital Anxiety and Depression Scale Anxiety/Depression; IEQ, Injustice Experience Questionnaire; LOT-R, Life Orientation Test-Revised;

BC, breast cancer; SN, sentinel node procedure; ALND axillary lymph node dissection

* Expected postsurgical pain ≥ 7

**HADS-A ≥ 8

*** HADS-D ≥ 7

groups differed in their experience with surgery. Patients in the low, compared to moderate fear group had undergone significantly more non-breast surgeries in the past. Moreover, patients in the moderate, compared to high fear group were significantly more likely to have had breast surgery previously. Finally, significant differences among all three fear groups were observed for expected postsurgical pain, as well as all investigated psychological questionnaires, but the one about perceived social support. Only the difference between depression scores of the high and moderate fear groups did not reach significance in the post-hoc tests.

## Overlap between high fear of surgery and psychological constructs with validated clinical cutoff-scores

Fig 4a displays the overlap among high fear of surgery, anxiety, and depression. Of patients with high fear, 26.0% (n = 13) exhibited combined anxiety and depression. All highly feared individuals symptomatic for depression, also had anxiety. Fig 4b shows, that 30.0% (n = 15) of participants with high fear of surgery, also suffered from anxiety and expected severe postsurgical pain.

## Sensitivity analyses

Participants who were eligible but for one of several reasons not included in the present analyses (n = 126), did not differ from the final sample with respect to their age, BC diagnosis and treatment (S2 Table). Further, results of the comparisons among fear groups proved to be robust even when using a more stringent cutoff for the transition from moderate to high fear of surgery (S3 Table).

## Discussion

This study investigated fear of surgery in a large sample of female BC patients awaiting surgical treatment at Norway's largest department for Breast Surgery. In line with prior expectations, the present results demonstrate that varying levels of fear of surgery are highly prevalent among this patient group. Only 1.5% of patients indicated to not experience any fear with respect to short- and long-term aspects of their upcoming treatment. This represents a novel insight, as the commonness of fear of surgery has neither been addressed in prior studies on female BC patients, nor in those on other surgical samples [21, 22, 24, 28–31, 50–59]. In view

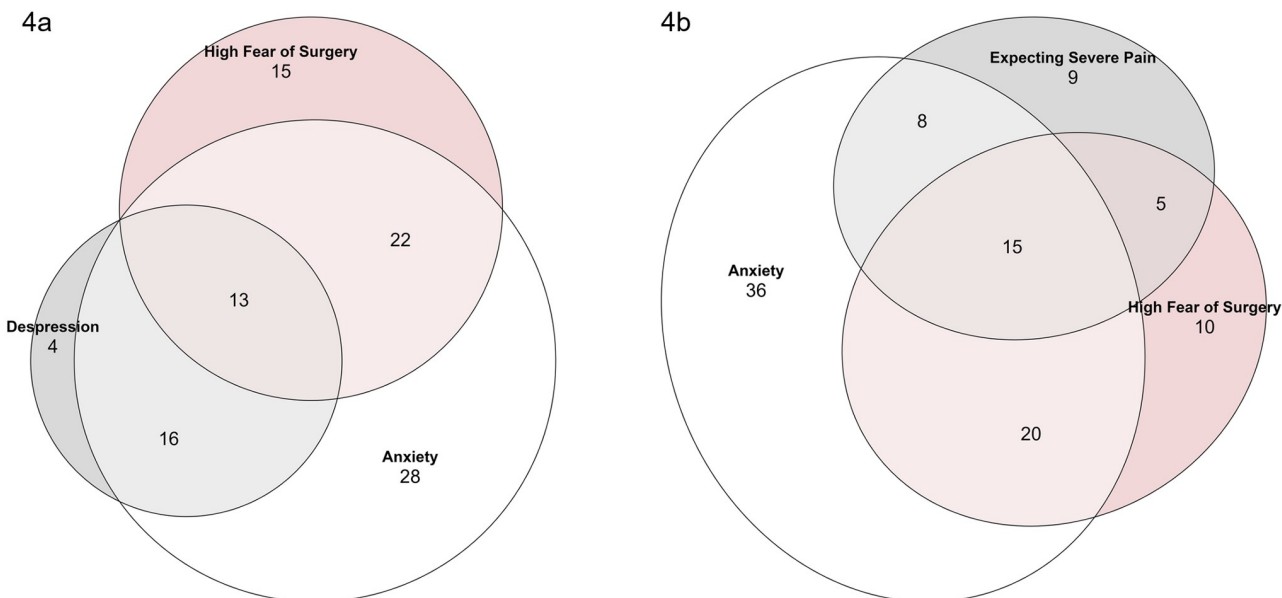

**Fig 4. Venn diagrams displaying the overlap among high fear of surgery, symptoms of anxiety and depression and the expectance of severe postoperative pain.**

of the adverse health-related postsurgical consequences that have been associated with fear of surgery in prior studies, describing the prevalence is relevant as it informs about the proportion of patients potentially being affected by these to some extent [25–28]. The average SFQ-score was 26.42 in the present sample. This is lower than hypothesized and below the average observed in a recent study which also assessed the SFQ in BC patients. Importantly, the average of the present study was even lower than that of the intervention group of that study (M = 28.95), which had received a 60-minute-long intervention aimed at reducing fear of surgery before SFQ-assessment [24]. Also, compared to samples of patients scheduled for various elective surgical procedures (M = 30.1–39.7), the average SFQ-score found here appears to be low [31, 50, 57]. One reason for these findings could be differences among health-care systems. In Norway, fixed 'cancer patient pathways' ensure that BC patients, except those needing neoadjuvant treatment, are operated within two weeks after the diagnosis. Above this, patients are offered consultations with specialist nurses trained to attend to their informational and psychological needs [60, 61]. Such care is reflected in high satisfaction rates among breast cancer patients [62]. Comparable pathways and high satisfaction are neither a worldwide, nor European standard in BC treatment [63]. Also, the setting of BC care is very different from that of elective surgery. Elective surgery is usually preceded by longer periods of suffering and waiting, and consultations are shorter. These are all factors that might influence fear ratings. Altogether fear of surgery was thus low, but omnipresent among in the present sample.

Within the frame of the SFQ, potential sources of fear of surgery were investigated. Findings demonstrate that women were slightly, but not significantly, more afraid of the short-, compared to long-term aspects of their surgery. This result corresponds well with results of previous studies employing the SFQ in BC [24] and elective surgery patients [28, 52–54, 57, 59]. Interestingly, the average SFQ-L-score found here was much higher than that observed in other samples of females suffering from benign gynecological diseases [28, 29, 51, 54]. One reason contributing to such differences might be that the consequences of benign, compared to malignant diseases, are usually associated with less uncertainty about the future. The

comparatively high SFQ-L score is relevant because these fears may persist beyond the surgery, thereby potentially taking a greater impact on patients' overall recovery. This assumption is also supported by prior studies [27–29, 52]. Lastly, among the individual items of the SFQ, women indicated to be most afraid of the surgery itself. This is consistent with prior hypotheses and previous qualitative reports [22]. Contrasting prior studies, however, was the finding that fear of the anesthesia was ranked second lowest. It might be, that when confronted with a fixed list of eight fear sources, as is the case for the SFQ, the anesthesia is feared, but relatively less than other aspects. Mean scores of individual items are rarely reported by prior studies and among the few stating them no clear pattern arises [50, 56–58]. Accordingly, between-study comparisons of SFQ item scores remain difficult.

Finally, the sample was divided into three groups (high, moderate, and low fear) and compared on a range of targeted outcomes. Irrespective of their current diagnosis and treatments, medical history, or sociodemographic factors, the higher a patients' fear group, the poorer their psychological constitution at the preoperative assessment was. More specifically, throughout low to high fear groups, patients exhibited increasing levels of anxiety, depression, and pessimism. Further, they experienced substantially more injustice related to their diagnosis and were significantly more likely to expect severe postsurgical pain. Importantly, these are all factors known to adversely affect the postoperative recovery of female BC patients [64–69] (for an additional overview see S1 Fig). Differences between groups on psychological measures were as expected. These results extend previous research in important respects: First, they are the first to highlight substantial differences in the psychological constitution of BC patients experiencing different degrees of fear of surgery. Second, they imply that presurgical levels of fear of surgery in this patient group might serve as an important prognostic marker. Lastly, they suggest that investigating fear of surgery on group, instead of sample level might be of added clinical value.

Generally, the average scores on psychological measures that were observed in the frame of the present study compare well to those reported in prior studies of presurgical BC patients [70–72]. When comparing mean values more closely, however, an interesting pattern emerged. With respect to all psychological outcomes investigated, the high fear group scored on average above, the moderate fear group within and the low fear group below a given mean score reported elsewhere [70–72]. With respect to anxiety and depression this pattern implied that proportions of clinically relevant symptoms within the low fear group mirrored that of the female Norwegian population [73], whereas those of the moderate group paralleled the average BC population [70–72] and the high fear group clearly exceeded both. This, irrespective of whether a HADS-D cutoff of 7 or 8 was used to indicate clinically relevant depression, which is an important remark as 8 was the cutoff used by all prior studies. Altogether, these comparisons emphasize the implications of belonging to either fear group and support hypotheses about patients' postsurgical recovery to differ depending on their SFQ-score. Moreover, given prior evidence highlighting that especially patients scoring high on the SFQ would welcome additional treatment for their fear, it appears likely that the need for and benefit from such treatment would also differ between fear groups [31].

The significant overlap among high fear of surgery, anxiety and depression symptoms, and the expectance of severe postsurgical pain highlights a high prevalence of psychological comorbidity in the high fear group. This constitutes yet another strong argument to assume that the prognosis of especially this group is likely to be impaired.

Interestingly, perceived social support did not differ significantly between groups. The same holding true for marital status and being a parent, which can also be viewed as aspects of social support. Given the great overlap among anxiety and high fear and considering results from prior studies demonstrating a negative relationship between social support and

preoperative anxiety, one would have expected that more feared patients would also perceive less social support to be available to them [58, 74–77]. This, however, was not the case, which is consistent with prior studies on fear of surgery [31, 78].

Lastly, differences between groups with respect to sociodemographic and investigated prior and current medical outcomes were fewer than expected. The effect of age was only significant for the difference between the low and moderate fear groups. Accordingly, conclusions about patients in a higher fear group generally being younger are not justified. This is consistent with prior work failing to demonstrate an association between age and fear of surgery [31]. Further, given prior results one would have expected that groups would differ in terms of assigned surgical procedure or psychological vulnerability [21]. Although such differences could not be shown here, proportions of individuals affected by any of these outcomes still increased linearly across fear groups. Given the small number of patients affected by any of these outcomes, future studies should aim to repeat these analyses in larger samples thereby empowering analyses.

## Strengths and limitations

Strengths of the present study are its broad inclusion criteria and the large sample lending it high external validity. Further to mention, the use of multiple imputation techniques to optimize statistical power. Above this, with respect to BC diagnoses and treatment the present sample mirrored the Norwegian BC population well [62]. Moreover, fear of surgery was measured using a validated questionnaire and groups were compared on a large variety of relevant sociodemographic and clinical outcomes. Consequently, compared to prior studies, a much more elaborate overview of the relevance of fear of surgery among this patient group could be achieved. Finally, to the best of our knowledge, the present study is the first to validate and apply a Norwegian version of the SFQ.

Factors limiting generalizability, however, are the following: First, since decreasing work-disability was a major goal of the PREVENT trial, patients older than 70 years were excluded. Accordingly, conclusions drawn based on the present results do not generalize to these patients. Second, participation in the PREVENT trial necessitated an interest in hypnosis, iACT and mindfulness, which might have introduced selection bias. Even though participants and eligible non-participants did not differ in their age or cancer-related variables, psychological differences cannot be excluded. Third, conclusions of the present work are confined to female BC patients of Northern European descent. As health-care systems, cultural stigma and knowledge about BC surgery differ greatly globally, it is likely that fear of surgery will do so as well. Another criticism might be that the time between presurgical assessment and surgery differed among patients. The impact of such differences, however, was regarded negligible given prior results showing that SFQ-ratings assessed at three different moments during the week before surgery differed only marginally [31]. Further, power was not calculated for the present study. A priori calculations were not possible, because participants were not primarily sampled for the present study and it was refrained from reporting post-hoc power as this only would have mirrored presented p-values [79–81]. Nevertheless, given the large group sizes, power should still have sufficed. A last limitation is the cross-sectional design of the present study. Consequently, a prognostic value of the fear groups can be assumed, but not confirmed based on the present results.

## Implications for future research

In the future, the follow-up data of the PREVENT trial should be used to investigate whether belonging to one of the three fear groups can predict patients' postoperative adaptation.

Further, as hypnosis and iACT exhibit the potential to mitigate fear, as well as other intra- and postoperative psychological outcomes, it would be interesting to explore whether fear of surgery moderates the effects of one or both interventions. Moreover, such a follow-up would bear the advantage that a receiver-operator-characteristic-curve could be employed to optimize the prognostic value of SFQ-cutoffs. Furthermore, fear is a stress response that is characterized by typical changes in the levels of certain stress-related biomarkers (cortisol, proinflammatory cytokines) [82]. As such data are available for the present sample, stress-biomarker differences between groups and time-related changes herein could also be investigated in the future. Finally, delay or refusal of BC treatment are especially on the African continent still prevalent [83]. Future studies should investigate in how far fear of surgery can account for this.

## Conclusion

Fear of surgery is a prevalent and relevant phenomenon among female BC patients. Of special concern are BC patients' comparatively high fears about long-term aspects of their surgery. Moreover, results highlight that the higher the fear group a patient scores in, the worse their presurgical psychological constitution. Such differences in mental health can be assumed to affect peri- and postsurgical outcomes. Taken together, the present results hint towards a potential role of fear of surgery as a prognostic marker for the postoperative recovery of female BC patients. If so, a routine clinical assessment of the SFQ might be of added value. A respective patients' fear group might guide estimates of patients' need for and benefit from additional pre- and postsurgical treatment. Further, fear of surgery might cater as a viable target for such interventions. Moreover, the ranking of individual SFQ-items presented here could inform the design thereof. Considering these possibilities, future studies are advised to focus on evaluating the prognostic value of the fear groups as presented here, as well as the need for and effect of targeted treatment.

## Supporting information

**S1 File. Validation of the Norwegian version of the Surgical Fear Questionnaire.**
(PDF)

**S2 File. R syntax of conducted analyses.**
(PDF)

**S1 Table. One-way ANOVA's assessing differences in mean scores of the Surgical Fear Questionnaire (SFQ) between fear groups (high, medium, low).**
(PDF)

**S2 Table. Sensitivity analyses: Comparison of eligible patients—Participating versus non-participating.**
(PDF)

**S3 Table. Sensitivity analyses: Comparison of two different cutoff scores to indicate high fear of surgery.**
(PDF)

**S1 Fig. Associations of fear of surgery and investigated psychological outcome measures with adverse health-related postsurgical consequences that were established by prior studies.**
(TIF)

## Acknowledgments

We would like to thank Martijn Smits for his support with the multiple imputation procedure and construction of Venn Diagrams.

## Author Contributions

**Conceptualization:** Sophia Engel, Silje Endresen Reme.

**Formal analysis:** Sophia Engel.

**Funding acquisition:** Henrik Børsting Jacobsen, Silje Endresen Reme.

**Methodology:** Sophia Engel, Henrik Børsting Jacobsen, Silje Endresen Reme.

**Project administration:** Henrik Børsting Jacobsen, Silje Endresen Reme.

**Resources:** Henrik Børsting Jacobsen, Silje Endresen Reme.

**Supervision:** Henrik Børsting Jacobsen, Silje Endresen Reme.

**Visualization:** Sophia Engel.

**Writing – original draft:** Sophia Engel.

**Writing – review & editing:** Sophia Engel, Henrik Børsting Jacobsen, Silje Endresen Reme.

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
