## [Decision Letter · Decision Letter 0]

9 Feb 2023

PONE-D-22-20334A cross-sectional study of fear of surgery in female breast cancer patients: Prevalence, severity, and sources, as well as relevant differences among patients experiencing high, moderate, and low fear of surgeryPLOS ONE

Dear Dr. Engel,

Thank you for submitting your manuscript to PLOS ONE. After careful consideration, we feel that it has merit but does not fully meet PLOS ONE’s publication criteria as it currently stands. Therefore, we invite you to submit a revised version of the manuscript that addresses the points raised during the review process.

As noticed by the reviewers, there are important changes that need to be applied before the manuscript can be considered for publication, such as The instrument to assess pain is specific for fibromyalgia, therefore why it is used on this study? also all the manuscript should be coherent between the sections, the reviewers have identified a lack of connection between the method-results and the introduction.

We look forward to receiving your revised manuscript.

Kind regards,

Alejandro Dominguez-Rodriguez, Ph.D.

Academic Editor

PLOS ONE

Journal Requirements:

Additional Editor Comments (if provided):

Dear Sophia Engel,

I have now the evaluation reports of 3 reviewers. Please proceed to review these comments and provide a response to the reviewers and the necessary modifications.

Thanks and best regards,

Dr. Alejandro Domínguez Rodríguez

Assistant Professor

University of Twente

Reviewers' comments:

Reviewer's Responses to Questions

**Comments to the Author**

1. Is the manuscript technically sound, and do the data support the conclusions?

Reviewer #1: Yes

Reviewer #2: Partly

Reviewer #3: Yes

2. Has the statistical analysis been performed appropriately and rigorously? 

Reviewer #1: Yes

Reviewer #2: Yes

Reviewer #3: Yes

3. Have the authors made all data underlying the findings in their manuscript fully available?

Reviewer #1: Yes

Reviewer #2: No

Reviewer #3: No

4. Is the manuscript presented in an intelligible fashion and written in standard English?

Reviewer #1: Yes

Reviewer #2: Yes

Reviewer #3: Yes

5. Review Comments to the Author

Reviewer #1: Dear Author

Thank you for sharing your interesting study. Regarding the manuscript that you submitted there are some points that should be considered:

1- Abstract: Please focus the abstract on your study and your results, Provide in the abstract an informative and balanced summary of what was done and what was found.

2- Introduction: Explain the scientific background and rationale for the investigation being reported

3- Methods: Explain how the study size was arrived, Present key elements of study design early in the paper, Describe any efforts to address potential sources of bias

4- In this study, you have used several questionnaires and items, How have you avoided the fatigue and overlaps of answering all the items at once for these participates in this certain situations?

Thank you

Reviewer #2: 1. This article is the result of a larger study consisting of a Trial. I think it would be more innovative if these results are presented as part of what was obtained in the trial.

2. Despite the fact that they made very clear and adequate statistics, the subject has been large studied, thus, lacks novelty. Even in the introduction they talk about other studies that have already studied the subject.

3. The main instrument used was not translated and they do not present a validation procedure or reliability data. The fact that it is not validated also means that there are no cut-off points established in the Norwegian population, despite the procedure they carried out to obtain them. In addition, I suggest to specify if it has been used in cancer patients, given the particularities of breast surgery.

4. I consider it important in the Outcome measures section to explicitly divide with primary outcomes and secondary outcomes sections.

5. The instrument to assess pain is specific for fibromyalgia. I consider that it is a limitation since it evaluates pain in parts of the body that is typical of fibromyalgia, leaving aside places in the body that patients with breast cancer may have pain.

6.It is not clear why the secondary outcomes were included: anxiety and depression, dispositional optimism, injustice experience and social support. Since in the introduction it is not specified.

7. It is excessive to present a figure with the results of each item of the SFQ. I suggest just describing it.

Reviewer #3: This is an interesting research report on the prevalence, severity, and sources of fear of surgery in breast cancer patients in Norway. The study is cross-sectional and includes 204 participants. I have some concerns, which I raise below. I hope the authors consider them as constructive and help them to improve their paper.

Abstract

Comment 1. Please indicate in brackets the country of the Hospital (especially useful for non-European readers).

Comment 2. I suggest including percentile 50 in the SFQ in the results section of the abstract.

Introduction

Comment 3. The authors state that the study is large-scale and the sample is representative of the BC population. However, the sample comes from a single-centre RCT and I have doubts about describing 200 participants as a large sample.

Methods

Comment 4. Line 208. Maybe there is an increased risk of type II error, but also of Type I. I suggest correcting for multiple testing since 23 between-group comparison analysis (ANOVA/chi square) are conducted. I suggest using the Benjamini-Hochberg correction (https://tools.carbocation.com/FDR). I think that the main significant differences will remain.

Results

Comment 5. Line 242. Please provide the median in text.

6. PLOS authors have the option to publish the peer review history of their article (what does this mean?). If published, this will include your full peer review and any attached files.

Reviewer #1: **Yes: **Sara Adimi

Reviewer #2: No

Reviewer #3: No

---

## [Author Response · Author response to Decision Letter 0]

28 Mar 2023

Dear all, 

Thank you so much for considering our manuscript. We attached a document called 'letter to the reviewers' where we answer to the comments of the reviewers we were provided with earlier. Please feel free to contact us if you should have any questions or if we should adapt something. 

We are already looking forward to be hearing back from you. 

Thank you so much already in advance! 

Sincerely, 

Sophia Engel

---

## [Decision Letter · Decision Letter 1]

4 Jun 2023

PONE-D-22-20334R1A cross-sectional study of fear of surgery in female breast cancer patients: Prevalence, severity, and sources, as well as relevant differences among patients experiencing high, moderate, and low fear of surgeryPLOS ONE

Dear Dr. Engel,

Thank you for submitting your manuscript to PLOS ONE. After careful consideration, we feel that it has merit but does not fully meet PLOS ONE’s publication criteria as it currently stands. Therefore, we invite you to submit a revised version of the manuscript that addresses the point of Reviewer 2 raised during the review process. Please submit your revised manuscript by Jul 19 2023 11:59PM. If you will need more time than this to complete your revisions, please reply to this message or contact the journal office at plosone@plos.org. Please include the following items when submitting your revised manuscript:A rebuttal letter that responds to each point raised by the academic editor and reviewer(s). You should upload this letter as a separate file labeled 'Response to Reviewers'.A marked-up copy of your manuscript that highlights changes made to the original version. You should upload this as a separate file labeled 'Revised Manuscript with Track Changes'.An unmarked version of your revised paper without tracked changes. You should upload this as a separate file labeled 'Manuscript'.If applicable, we recommend that you deposit your laboratory protocols in protocols.io to enhance the reproducibility of your results. Protocols.io assigns your protocol its own identifier (DOI) so that it can be cited independently in the future. For instructions see: https://journals.plos.org/plosone/s/submission-guidelines#loc-laboratory-protocols. Additionally, PLOS ONE offers an option for publishing peer-reviewed Lab Protocol articles, which describe protocols hosted on protocols.io. Read more information on sharing protocols at https://plos.org/protocols?utm_medium=editorial-email&utm_source=authorletters&utm_campaign=protocols.

We look forward to receiving your revised manuscript.

Kind regards,

Alejandro Dominguez-Rodriguez, Ph.D.

Academic Editor

PLOS ONE

Journal Requirements:

Reviewers' comments:

Reviewer's Responses to Questions

**Comments to the Author**

1. If the authors have adequately addressed your comments raised in a previous round of review and you feel that this manuscript is now acceptable for publication, you may indicate that here to bypass the “Comments to the Author” section, enter your conflict of interest statement in the “Confidential to Editor” section, and submit your "Accept" recommendation.

Reviewer #1: All comments have been addressed

Reviewer #2: All comments have been addressed

Reviewer #3: All comments have been addressed

2. Is the manuscript technically sound, and do the data support the conclusions?

Reviewer #1: Yes

Reviewer #2: Yes

Reviewer #3: Yes

3. Has the statistical analysis been performed appropriately and rigorously? 

Reviewer #1: Yes

Reviewer #2: Yes

Reviewer #3: Yes

4. Have the authors made all data underlying the findings in their manuscript fully available?

Reviewer #1: Yes

Reviewer #2: No

Reviewer #3: Yes

5. Is the manuscript presented in an intelligible fashion and written in standard English?

Reviewer #1: Yes

Reviewer #2: Yes

Reviewer #3: Yes

6. Review Comments to the Author

Reviewer #1: Dear Authors,

I would like to thank you for submitting the revised version of your manuscript.

Your revisions have addressed all the comments and feedback that I provided in my previous review.

Overall, this is a high quality and well-written manuscript

Reviewer #2: I consider that all comments were addressed. However, one last suggestion would be to explicitly state the objective of the study. I suggest add it before the hypotheses.

Reviewer #3: All commnents have been addressed. All commnents have been addressed. All commnents have been addressed.

7. PLOS authors have the option to publish the peer review history of their article (what does this mean?). If published, this will include your full peer review and any attached files.

Reviewer #1: No

Reviewer #2: **Yes: **Reyna Jazmín Martínez-Arriaga

Reviewer #3: No

---

## [Author Response · Author response to Decision Letter 1]

6 Jun 2023

Dear all, 

Thank you so much for considering our manuscript. We attached a document called 'letter to the reviewers' where we answer to the comments of the reviewers we were provided with earlier. Please feel free to contact us if you should have any questions or if we should adapt something. 

We are already looking forward to be hearing back from you. 

Thank you so much already in advance! 

Sincerely, 

Sophia Engel

---

## [Editor Report · Decision Letter 2]

12 Jun 2023

A cross-sectional study of fear of surgery in female breast cancer patients: Prevalence, severity, and sources, as well as relevant differences among patients experiencing high, moderate, and low fear of surgery

PONE-D-22-20334R2

Dear Dr. Engel,

We’re pleased to inform you that your manuscript has been judged scientifically suitable for publication and will be formally accepted for publication once it meets all outstanding technical requirements.

Kind regards,

Alejandro Dominguez-Rodriguez, Ph.D.

Academic Editor

PLOS ONE
---

## [Editor Report · Acceptance letter]

15 Jun 2023

PONE-D-22-20334R2 

A cross-sectional study of fear of surgery in female breast cancer patients: Prevalence, severity, and sources, as well as relevant differences among patients experiencing high, moderate, and low fear of surgery 

Dear Dr. Engel:

I'm pleased to inform you that your manuscript has been deemed suitable for publication in PLOS ONE. Congratulations! Your manuscript is now with our production department. 

Kind regards, 

on behalf of

Dr. Alejandro Dominguez-Rodriguez 

Academic Editor

PLOS ONE